# Optimization Methods Applied to Motion Planning of Unmanned Aerial Vehicles: A Review

**Amber Israr [1], Zain Anwar Ali [1,\*], Eman H. Alkhammash [2] and Jari Juhani Jussila [3]**

[1] Electronic Engineering Department, Sir Syed University of Engineering & Technology, Karachi 75300, Pakistan; aisrar@ssuet.edu.pk

[2] Department of Computer Science, College of Computers and Information Technology, Taif University, Taif 21944, Saudi Arabia; eman.kms@tu.edu.sa

[3] HAMK Design, Factory, Häme University of Applied Sciences, 13100 Hämeenlinna, Finland; jari.jussila@hamk.fi

[\*] Correspondence: zaali@ssuet.edu.pk

**Abstract:** A system that can fly off and touches down to execute particular tasks is a flying robot. Nowadays, these flying robots are capable of flying without human control and make decisions according to the situation with the help of onboard sensors and controllers. Among flying robots, Unmanned Aerial Vehicles (UAVs) are highly attractive and applicable for military and civilian purposes. These applications require motion planning of UAVs along with collision avoidance protocols to get better robustness and a faster convergence rate to meet the target. Further, the optimization algorithm improves the performance of the system and minimizes the convergence error. In this survey, diverse scholarly articles were gathered to highlight the motion planning for UAVs that use bio-inspired algorithms. This study will assist researchers in understanding the latest work done in the motion planning of UAVs through various optimization techniques. Moreover, this review presents the contributions and limitations of every article to show the effectiveness of the proposed work.

**Keywords:** unmanned aerial vehicle; motion planning; optimization techniques

## 1. Introduction

Flourishing high-tech innovations are making aerial robots an integral part of our daily lives. There are extensive research and analyses on flying robots that possess the mobility given by flight [1,2]. Among these, Unmanned Aerial vehicles (UAVs) are vastly used flying robots due to these distinguishing advantages over others, i.e., budget-friendly, small-sized, lighter in weight, and portable. Moreover, the state-of-the-art characteristics of UAVs are position controlling, sensor employment, auto-level application, structure monitoring, etc. [3–5]. It also has a diverse array of applications, whether in the military or civilian sectors [6]. There are two primary models of UAVs; one is fixed-wing, and the other one is multi-rotor UAVs. The essentials of UAV performance are higher in complex tasks or uncertain environments. Usually, a single UAV has a small size, which limits its volume of sensing, communication, and computation [7]. Thus, cooperative UAVs working together have more benefits and potential results in comparison to a single UAV [8]. A few of them are cost and operation time reduction, low failure of missions, and achievement of higher flexibility, survivability, configurability, and multi-tasks capability [9].

**Background:** It is one of the utmost evolving technologies from the 18th century and is advancing till now. At first, in 1849, Montgolfier's French brothers and Austrians employed unmanned balloons filled with bombs [10]. The development of UAVs with cam-

eras occurred in 1860, which helped with vigilance [11]. In 1917, Charles F. Kettering invented an Aerial Torpedo and named unmanned balloons bugs. The Royal Navy tested a radio-controlled pilotless aircraft during the 1930s [12]. The 1940s were marked by operation Aphrodite, in which a formation of UAVs with handheld control took place for the first time and radio control-based Queen Bee was developed. A few of them were Pioneer, Predator, Ryan fire bee, etc. In 2003, Amazon started using UAVs commercially [13].

**Related Work:** Extensive analysis of various core issues on UAVs related to motion planning under different circumstances and environments [14]. To design motion control protocols and select path planning techniques, many problems and factors require serious considerations [15]. Numerous researchers have proposed distributed consensus-based motion controls for results with efficacy and accuracy. Some developed leader-follower strategies for efficient outcomes [16]. Some analyses have used bio-inspired algorithms for better path planning with minimal run time. Many employed hybrid algorithms for optimal path planning and achieved a reduction in cost and convergence time [17].

**Motivation and Contribution:** The motivation for this paper is to assemble various strategies used in different research together in a single place. This will help researchers select the best strategy for their required missions while comparing the explorations and exploitations of all the strategies. To overcome the hurdles of different limitations, uncertain disturbances, and complexities, appropriate strategies are essential. This makes the system more stable and efficient and reduces the convergence rate and cost. The prime contributions of this review paper are:

A. The evaluation of the challenges faced by UAVs under different scenarios.
B. Summarizing various promising motion planning techniques and algorithms for determining the optimum path for UAVs.
C. To gather the contributions and limitations presented in each article.

This review is based on the research studies and publications from reputed authors in the field of motion planning techniques used for UAVs over the last three years.

**Organization:** The layout of this paper has many sections, of which Section 2 discusses the challenges that a UAV faces. Section 3 reviews recent developments in motion control and path planning mechanisms. Section 4 evaluates the motion planning and optimization algorithms. Section 5 presents the discussion. Section 6 provides the conclusion, and Section 7 gives directions for future work.

## 2. Challenges in Unmanned Aerial Vehicles

There are extensive investigations regarding UAVs, but still, they face various challenges. The prime challenges that all the researchers face include the selection of UAVs with appropriate path planning that is suitable for the mission [18]. Then, forming efficient motion control and achieves optimal path planning. Moreover, employing proper techniques for navigation and communication so that obstacle avoidance and collision avoidance are possible. Along with this certification, regulation and human-machine interface issues are of much importance. Below are some of the challenges that require serious consideration:

### 2.1. Navigation and Guidance

UAVs have to track their mobility by measuring their distances, making maps, and sensing physical surroundings. To determine the positions of aerial robots, it is essential to develop a navigation system, which is automatic and does not require human interventions [19]. These robots are for flying at higher altitudes and under different environments and hazards. Therefore, the safety and reliability of the system to operate properly are major challenges.

*2.2. Obstacle Detection and Avoidance*

The navigation of UAVs is much influenced by obstacles and collisions. Providing UAVs with an ideal environment is not a viable option. Obstacles that come in the path can be avoided. Moreover, the performances of multiple aerial robots are more beneficial and efficient than a single flying robot. Working in groups can result in collisions. UAVs must be furnished with algorithms or techniques that can handle these issues [20].

*2.3. Shape and Size*

Nowadays, UAVs are widely used for different purposes. They are required to fly at different levels with different ranges. Some have to stay for a longer period to accomplish their missions. Some use runways for flying and landing. Some have to pass through narrow areas. To solve all these issues, it is necessary to consider the appropriate shapes and sizes of UAVs according to the missions [21]. Figure 1 shows some of these challenges faced by UAV [22].

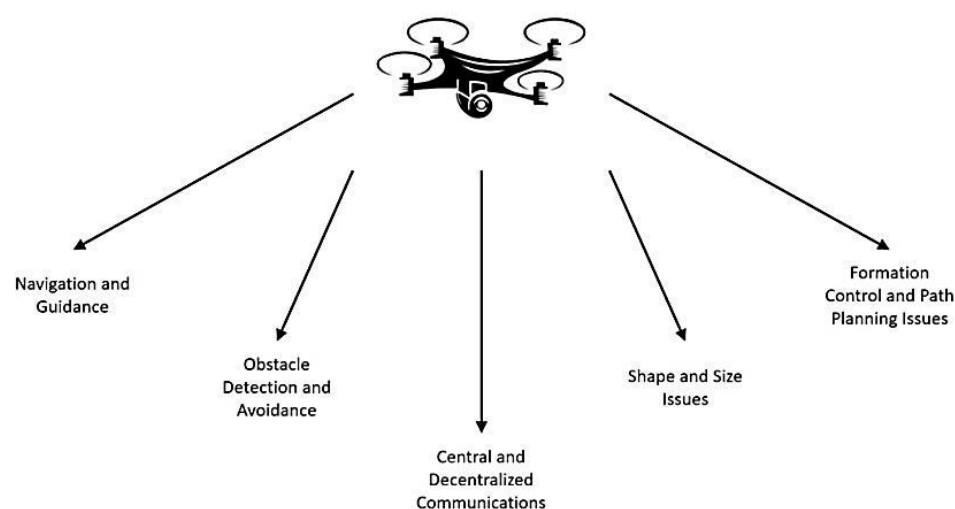

**Figure 1.** Challenges in UAVs [22].

*2.4. Formation Control and Path Planning Issues*

2.4.1. Formation Control Issues

There are numerous studies on motion control, but it still lacks and requires consideration and further handling. For example, there stands a need to tackle distributed levels with their effects properly. Similarly, machine learning and reinforcement learning require a longer time for the online learning period and huge data sets for offline training. Therefore, the integration of artificial intelligence (AI) techniques into control protocols is essential. One more challenge in motion control protocol is its robustness, which is highly influenced by environmental disturbances [23].

2.4.2. Path Planning Issues

Path planning is to obtain a path for UAVs from the starting to the goal point in such a way that they will carry out their tasks efficiently. UAVs require optimal paths that satisfy their performance constraints and ensure collision avoidance. Such optimal and dynamic paths consume less time and energy. Path planning is a global optimization problem that requires various technologies and algorithms to be integrated [24].

Among all the challenges, the most crucial is path planning and motion control for UAVs. These require considerations so that the UAV can perform well during tasks under

any environmental conditions. Several research centers, academies, and industries are analyzing the aforementioned challenges and trying to overcome these issues by developing more improved strategies. Section 3 reviews the development of various protocols and techniques used for the above challenges.

## 3. Recent Developments in UAVs

UAV technology is expanding due to technological innovations. UAVs are becoming more affordable and easy to use, which enhances their application in diverse areas [6]. This paper reviews the strength and development of navigation, communication, shape and size, collision avoidance, motion control methods, and path planning techniques. It deliberates how they provide solutions to challenging problems while making a considerable impact.

### 3.1. Developments in Navigation and Guidance of UAVs

Navigation technology is quite significant for UAV flight control. Various developed navigation technologies possess different features. Such as satellite, geometric, integrated, Doppler, and inertial navigations. Different purposes require different navigation technologies. The main navigation systems for UAVs are a tactical or medium range navigation system and a high-altitude long-endurance navigation system [25,26]. Development in navigation can be evaluated as:

D.　**High-performance Navigation with Data Fusion:** Navigation uses a Kalman filter; China introduced a data fusion mechanism using this filtering technology. This data fusion is improved by using AI technology. It helps to determine the flight status and guarantees the normal flight of UAVs.

E.　**New Inertial Navigation System:** Many researchers rendered services to develop optical fiber inertial navigation and laser inertial navigation. Improvement was required by the industry. The widely used silicon micro resonant accelerometer helps in UAV navigation. It simplifies the weight and volume, consumes less energy, and refines flight pliability.

F.　**Intelligent Navigation Ability:** An emergency navigation system utilizes various adaptive technologies along with mission characteristics and modes. Moreover, information technology is applied to boost the UAV technology and upgrade the navigation system.

### 3.2. Developments in Shape and Size of UAVs

Earlier, UAVs were applicable for military purposes only, but now they are used for various tasks. This is all due to the rapid progress in developing UAVs with a wide range of shapes and sizes [27]. Different UAVs are utilized for different purposes. According to physical types, we have fixed-wing and multi-rotor UAVs.

**Fixed-Wing UAVs:** These UAVs possess only one long wing on any body's side and require a runway or a broad and flat area. These can consume less battery; therefore, they can stay in the air for maximum hours. They are widely used for long-distance purposes, especially for military surveillance.

**Multi-Rotor UAVs:** These UAVs are built up with multiple propellers and rotors and do not require a runway for vertical flying and landing. With more rotors, the position of UAVs can be controlled in a better way. Mostly quad-rotors are used for small and regular-sized UAVs. Similarly, UAVs are classified based on their sizes into micro or mini-UAVs, tactical UAVs, strategic UAVs, and special-task UAVs.

**Micro and Mini-UAVs:** Many missions require small UAVs. Such as surveillance inside buildings, Nuclear, Biological, and Chemical (NBC) sampling, the agricultural sector, and broadcast industries. Micro and mini-UAVs were developed for these purposes. The take-off weight of a micro-UAV is 0.1 kg, and a mini-UAV is less than 30 kg. Both fly below 300 m with less than 2 h of endurance. The communication range is up to 10 km.

**Tactical UAVs:** Missions such as search and rescue operations, mine detection, communication relays, and NBC sampling use tactical UAVs. They can have a take-off weight of up to 1500 kg. Tactical UAVs can fly up to 8000 m with an endurance of up to 48 h. The communication range is around 10–500 km.

**Strategic UAVs:** For airport security, communication relays, intercept vehicles, and RSTA, strategic UAVs are highly suitable. They can have a maximum take-off weight of around 12,500 kg. They can fly up to 20,000 m with 48 h of endurance. The communication range is more than 2000 km.

*3.3. Developments in Collision Avoidance of UAVs*

A collision usually occurs between a UAV and its neighboring UAV or an obstacle whenever there is less distance between them. A collision avoidance system (CAS) makes sure that no collision takes place with any stationary or moving obstacle [28]. The CAS first requires the perception phase and is then followed by the action phase.

**Perception Phase:** CAS detects an obstacle in this phase while utilizing various active or passive sensors according to their functionality principle. Active sensors possess their sources for wave emission or light transmission along with the receiver or detector. The most-used active sensors include radars, sonar, and LiDARs. All of these use minimum processing power, give a quick response, are less affected by weather, scan bigger portions in minimum time, and can return various parameters of the obstacles effectively. Whereas passive sensors are only capable of reading the emitted energy from another source such as the sun. Widely used passive sensors are visual or optical cameras and infrared (IR) or thermal cameras. The image formed by a visual camera requires visual light, whereas a thermal camera requires IR light.

**Action Phase:** This phase utilizes four prime strategies for collision avoidance. These are geometric, force-field, optimized, and sense and avoid methods. The geometric approach utilizes the information about the location and velocity of the UAV along with its obstacle or neighbors. This is performed by trajectory simulation in which nodes are reformed for collision avoidance. In force-field, the approach manipulates the attractive or repulsive forces to avoid collisions. In the optimized method, the parameters of obstacles, which are already known, are utilized for route optimization. In the sense and avoid technique, runtime decisions are made based on obstacle avoidance. The development in CAS helps in simple tasks by warning the vehicle operator and in complex tasks partially or completely controlling the system for collision avoidance.

*3.4. Developments in Formation Control Protocols of UAVs*

Formation control aims to generate control signals, which pilot UAVs to form a specific shape. Along with the architecture of motion control, the developed strategies for obtaining it are of much importance [29].

**Formation Control Design:** Motion controls of UAVs require a flow of information within its team; therefore, it uses communication architectures. Its architecture is categorized as:

There may be a lack of availability of global information in a single UAV for a whole operation. Due to its restricted capabilities to compute and communicate, centralized architecture is considered or used rarely. Decentralized architecture is preferred more for multi-UAV systems and uses the consensus algorithm technique for designing it. It is based on local interactions with the neighbors while maintaining a certain distance.

**Formation Control Strategies:** Various developed control approaches are discussed here that aid the researchers and possess certain benefits and limitations. They are:

i. **Leader-Follower Strategy:** As obvious from its title, this approach assigns one UAV as a leader, while the remaining UAVs as followers in a group. The mission information remains with the leader only while the followers chase their leader with pre-designed spaces. The major benefit of this strategy is that it can be implemented

simply and easily. Due to leader dependency, this strategy faces single-point failures. This limitation can be compensated by assigning multi-leaders and virtual leaders.

ii.　**Behavior-based Strategy:** This approach produces control signals, which consider several mission essentials, by adding various vector functions. Its greatest merit is that it is highly adaptable to any unknown environment. Its demerit is the requirement to model it mathematically, which leads to difficulty in analyzing system stabilities.

iii.　**Virtual Structure Strategy:** This approach considers rigid structure for the desired shape of the group of UAVs. To achieve the desired shape, there is a need to fly each UAV towards its corresponding virtual node. Abilities to maintain the formation and fault-tolerance are its greatest advantages. This approach faces failure when the detection of a UAV is faulty in the formation. The compensation for this faulty UAV requires reconfiguration of the formation shape. This approach calls for a strong ability to compute, which is a disadvantage of this approach.

### 3.5. Developments in Path Planning Techniques of UAVs

Path planning aims to design a flight path towards a target with fewer chances of being demolished while facing limitations. Extensive research proposed different methods that overcome the path planning complexity of UAVs. To design algorithms for path planning, certain parameters, such as obstacles, the environment, and constraints, require selection with considerations [30]. The approaches employed for path planning have classifications based on their features and methodology.

## 4. Motion Planning and Optimization

### 4.1. Motion Planning

In robotics, motion planning refers to the act of dissolving a specified mobility goal into distinct motions. However, it is used to fulfill movement limitations while also potentially optimizing some components of the motion. However, motion planning is the challenge of planning for a vehicle that operates in areas with a high number of objects, performing actions to move through the environment as well as modify the configuration of the objects [31]. Even though the motion planning situation has arisen in continuous C-space, the calculation is discrete. As a result, we need a means to "discretize" the problem if we want an algorithmic solution. As a result, there are mainly two types of planning, combinatorial planning and sampling-based planning.

#### 4.1.1. Combinatorial Motion Planning

Combinatorial Motion Planning is a type of motion planning that involves more than one approach to achieve the task, as shown in Figure 2. Although combinatorial motion planning discovers the pathways through the continuous configuration space, by using these strategies, researchers obtain a better result. The effective combination of algorithms is commonly based on bio-inspired algorithms with different approaches.

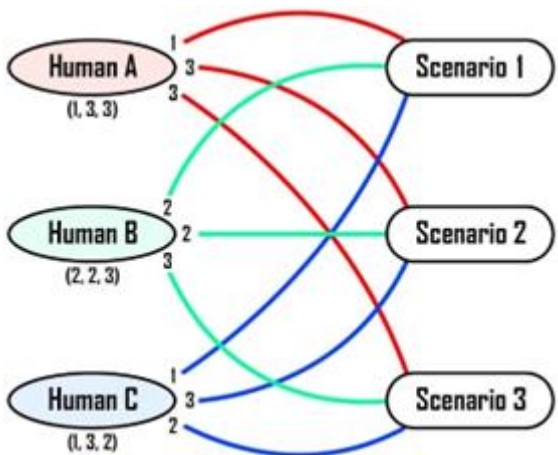

**Figure 2.** Three tasks Combinatorial Optimization example [32].

### 4.1.2. Sampling-Based Motion Planning

Random selection is used in sampling-based motion planning to build a graph or tree (path) in C-space on which queries (start/goal configurations) can be solved, as shown in Figure 3. To increase planner performance, we look at a variety of general-purpose strategies. At times over the past years, sampling-based path planning algorithms, such as Probabilistic Road Maps (PRM) and Rapidly Exploring Random Trees (RRT), have been demonstrated to perform effectively in reality and to provide theoretical assurances such as probabilistic completeness.

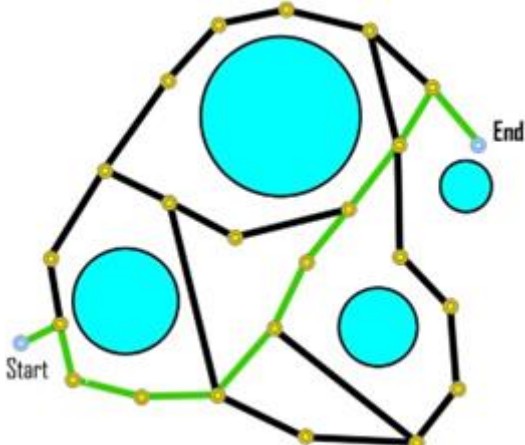

**Figure 3.** Sampling-based motion planning in the complex environment [33].

### 4.2. Optimization Approach in Motion Planning.

The world has a desire for optimization concerning every natural phenomenon and its aspects. Therefore, many researchers developed optimization methods for multi-dimensional problems in various areas. These algorithms provide optimum solutions to the motion planning problems of UAVs, such as reducing production costs, convergence rate, energy consumption, and enhancing strength, efficiency, and reliability. The optimization algorithms are classified into biological algorithms, physical algorithms, and geographical algorithms, as presented in Figure 4 [34,35]. Biological algorithms have further classifications, namely swarm-based and evolution-based algorithms.

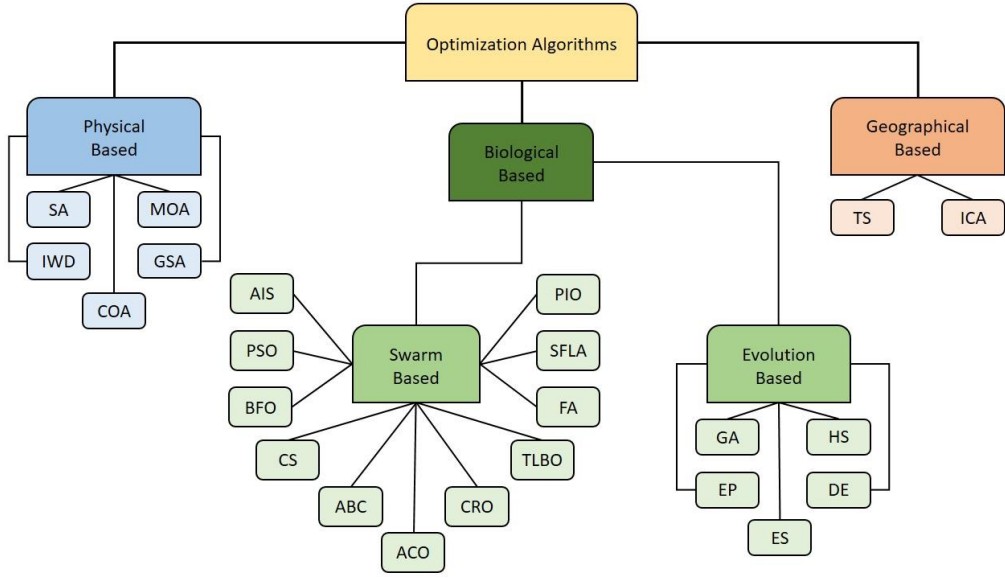

**Figure 4.** Classification of Optimization algorithms [36].

Biological Algorithms

Bionic researchers on a natural pattern developed nature-based algorithms and termed them biological algorithms. These are stemmed according to the correspondence between biological evolution and activities. The prime benefit of biological algorithms is their strength to tackle static as well as dynamic threats and ensure offline working. Without classifying these algorithms into further groups, we can label them as memetic algorithms. On the contrary, we can classify these algorithms into two categories, evolution-based algorithms and swarm-based algorithms [37].

A.  Evolution-Based Algorithms

An evolution-based algorithm provides an optimal path for UAVs with consideration of three aspects. These aspects include travel distance, cost incurred, and path reliability cost to track that path. These evolutionary algorithms choose practical and achievable solutions randomly as the first generation and consider the parameters later to explain which randomly selected feasible solutions are appropriate or not. For determining curved paths with essential aspects in 3D terrain; an offline path planner with an evolutionary algorithm is required [38]. By taking aspects into account, for example, beeline to destination, min-max distance related to targets, and topographical obstacles free tracks, one can display the B-spline curve as a flying path. Some examples of these algorithms include Genetic algorithm (GA), Evolutionary Programming (EP), Evolutionary Strategy (ES), Differential Evolution algorithm (DE), and Harmony Search algorithm (HS).

GA gives the best optimal results in search space using three steps selection, crossover, and mutation. Besides its benefits, sometimes it gives long and premature convergence and loses optimal results. Moreover, it is not applied to real-time data. In 1990, Fogel introduced a technique called EP. It reaches optimal results after many iterations. Similarly, another evolutionary algorithm is ES, which uses specified principles in optimization problems. DE employs real coding instead of binary coding. It refines the final path while reducing the computational cost. The evolutionary algorithm that mimics a musician's improvisation process is the HS algorithm. It shows promising results in optimization problems. It is further improved with various versions.

B.  Swarm-Based Algorithms

Nature-based along with population-based algorithms evolved into swarm-based algorithms [39]. The swarm represents the combined behavior of all the agents. Agents in a swarm have limited capabilities, but working together, they achieve the given tasks while being at distances. As a result of which, fast, low cost, and optimal solutions are obtained

even in the uncertainties and complexities. Some examples of these algorithms include Artificial Immune System (AIS), Particle Swarm Optimization (PSO), Bacteria Foraging Optimization algorithm (BFO), Cuckoo Search algorithm (CS), Artificial Bee Colony algorithm (ABC), Ant Colony Optimization algorithm (ACO), Coral Reef Optimization algorithm (CRO), Teaching–Learning Based Optimization algorithm (TLBO), Firefly algorithm (FA), Shuffled Frog Leaping algorithm (SFLA), and Pigeon-Inspired Optimization (PIO).

AIS is an intelligent swarm-based algorithm that is modeled on the natural principles of the immune system of humans. It has the characteristics of the immune system of memory and learning to utilize for solving problems. It gives adequate trajectories in path planning with less computation. The development of PSO is based on the mobility theory of an insect crowd. In the layout of this fact-finding approach, every solo particle in the crowd recognizes the points given by the last swarm and produces a velocity vector towards the target point. The key benefit of this algorithm is that it is capable of obtaining optimal path planning in 3D, whereas its disadvantages are premature convergence and high time complexity. Passino introduced an algorithm based on the foraging behavior of Escherichia coli bacteria that lies in human intestines. He labeled this intelligent algorithm as BFO.

It provides rapid convergence and a global search. The CS algorithm replaces the average solutions and applies the solution that is potentially better. The ABC algorithm provides solutions to various optimization problems having constraints. The ACO algorithm is based on depositing characteristics of ants during food search and proved to be a meta-heuristic technique to derive the shortest path while dealing with continuous and multi-objective path planning issues. The CRO algorithm works efficiently with many advantages for difficult optimization problems. The TLBO algorithm requires minimum computational memory and can be employed easily. FA works efficiently for multimodal optimization problems. It finds the best location for UAVs with less energy consumption. SFLA depends on frogs' clusters that are looking for food. It gathers the best frog, which can give local optimum and evolves the frog with inaccurate positions. It continues making iterations until the accomplishment of an optimal path with better convergence. PIO works via sharing information and striving among all to quickly achieve the optimal global solution.

C. Physical Algorithms

Heuristic algorithms that imitate physical laws and processes of nature are known as physical algorithms. These algorithms copy the physical conduct and characteristics of matter [40]. These are applicable for non-linear, high-dimensional, multimodal as well as complex optimization problems. There is very little research available on physical algorithms. These are categorized as Simulated Annealing (SA), Gravitational Search algorithm (GSA), Chaotic Optimization algorithm (COA), Intelligent Water Drops algorithm (IWD), and Magnetic Optimization algorithm (MOA). SA is suggested after a technique, annealing in metallurgy. It is employed for more complex computational optimization problems and gives approximate global optimum within a fixed time. GSA is a newly introduced algorithm that mimics laws of motion and gravitational law. It is applied to optimization problems with various functions. COA is an easily implemented and powerful mechanism that can escape convergence to a local optimum within a short time. The IWD algorithm is based on how natural rivers can find the best paths among many probable paths to their ultimate destination. MOA, a newly emerging algorithm, is derived from the basic principles of magnetism. The dual function of this algorithm can balance the disadvantages against the advantages in optimization problems.

D. Geographical Algorithms

The meta-heuristic algorithms that give random outcomes in geographical search space are labeled as geographical algorithms [41]. Some of the geographical algorithms are the Tabu Search algorithm (TS) and Imperialistic Competition algorithm (ICA). The

TS algorithm determines an optimal solution among various feasible solutions. Its memory can recall the recent optimal solution and guide the search to trace the previous solutions. It is employed for optimization problems in various areas. Another geographical algorithm for the global best solution in optimization is ICA. It imitates sociopolitical imperialist competition. It involves imperialistic competition among empires along with assimilation and revolution of colonies and so on. Due to its robust searching ability, it provides many benefits in optimization problems.

Among all the aforementioned algorithms, most are based on the swarm. These population-based algorithms are robust at obtaining better global solutions via their cooperative and self-adaptive abilities. These algorithms are employed for solving challenging issues of UAVs. This review paper gives details on a comparison of the aforesaid algorithms used for motion control and path planning of UAVs.

## 5. Related Review

To succeed, most motion planning approaches necessitate the use of appropriate optimization algorithms. These strategies can be used on a single UAV as well as a group of UAVs or a swarm of UAVs. When several UAV missions are viable for civilian objectives, a nature-inspired algorithm is required for control and optimization. Table 1 presents a detailed overview of the manuscripts related to motion planning problems of UAVs. The review also helps scholars with the optimization techniques applied to single or multiple UAVs.

**Table 1.** A detailed study on the Motion Planning of UAVs using the optimization approach.

| Ref | Topic | Optimization Approach | UAV Type | Contributions | Limitations |
|---|---|---|---|---|---|
| [42] | "Collision free 4D path planning for multiple UAVs based on spatial refined voting mechanism and PSO approach" | PSO | Multiple | ▪ Enhances searching ability and improves velocity. <br> ▪ Gives collision-free paths. | ▪ Returns to initial points in extreme conditions. |
| [43] | "Dynamic Discrete Pigeon-inspired Optimization for Multi-UAV Cooperative Search-attack Mission Planning" | D²PIO | Multiple | ▪ Ability to switch task. <br> ▪ Superior performance in discrete environment. | ▪ Frequent switching led to incomplete tasks. <br> ▪ Computational cost is higher due to population size. |
| [44] | "MVO-Based Path Planning Scheme with Coordination of UAVs in 3-D Environment" | MA | Multiple | ▪ Gives optimized path costs. <br> ▪ Maintains coordination. | ▪ Do not give dynamic obstacles. <br> ▪ Does not consider hardware-oriented constraints. |
| [45] | "UAV trajectory optimization for Minimum Time Search with communication constraints and collision avoidance" | ACO | Single | ▪ Detects the target quickly. <br> ▪ Maintains connection with GCS and avoids collision. | ▪ Greater computational time. <br> ▪ A mandatory ground connection is needed to obtain desired results. |
| [46] | "Efficient path planning for UAV formation via comprehensively improved particle swarm optimization" | IPSO | Multiple | ▪ Boosts the convergence rate. <br> ▪ Improves the solution optimality. | ▪ Does not allow path re-planning with moving and unexpected obstacles. |

| | | | | Advantages | | Limitations |
|---|---|---|---|---|---|---|
| [47] | "Secrecy improvement via a joint optimization of UAV relay flight path and transmit power" | PSO | Single | ▪ Enhances the secrecy capacity.<br>▪ Allows optimum position flying. | ▪ | Needs further improvement for full-duplex relaying. |
| [48] | "Trajectory Planning for UAV Based on Improved ACO Algorithm" | MACO | Multiple | ▪ Optimized initial trajectory<br>▪ Proposed trajectory correction schemes for collision avoidance. | ▪ | No real-time trajectory planning used. |
| [49] | "Optimized Path-Planning in Continuous Spaces for Unmanned Aerial Vehicles Using Meta-Heuristics" | DE PSO GA | Multiple | ▪ Less computation for first feasible path.<br>▪ DE overtakes PSO and GA in convergence. | ▪ Work exists for static environment only.<br>▪ No real-time implementation. |
| [50] | "Multi-UAVs trajectory and mission cooperative planning based on the Markov model" | SA | Multiple | ▪ Improves drone survivability.<br>▪ Solves multi-aircraft mission planning problems. | ▪ | Needs NP problem exploration. |
| [51] | "PSO-based Minimum-time Motion Planning for Multiple Vehicles Under Acceleration and Velocity Limitations" | PSO | Multiple | ▪ Minimizes the travelling time for slowest UAV.<br>▪ Reduces the parameters for mathematical modeling. | ▪ No control law for motion tracking.<br>▪ Only applied to selected vehicles. |
| [52] | "Information fusion estimation-based path following control of quad-rotor UAVs subjected to Gaussian random disturbance" | GIFC | Single | ▪ Reduces the design complexity.<br>▪ Allows trajectory tracking with high accuracy. | ▪ Contains a huge amount of matrix inversion operations. |
| [53] | "3D multi-UAV cooperative velocity-aware motion planning" | A* | Multiple | ▪ Shows a higher possibility of reaching destinations.<br>▪ Reduces time costs and paths. | ▪ Does not serve complex missions and more UAVs. |
| [54] | "Unmanned aerial vehicle swarm distributed cooperation method based on situation awareness consensus and its information processing mechanism" | SDCM | Multiple | ▪ Works efficiently in a complex and antagonistic mission environment.<br>▪ Obtains the mission essentials at a bearable cost. | ▪ On a larger scale, communication topology and management mode changes. |
| [55] | "A co-optimal coverage path planning method for aerial scanning of complex structures" | CCPP PSO | Multiple | ▪ Optimizes path efficiency and inspection quality.<br>▪ Provides improved flexible options. | ▪ The exponential growth of complexity occurs as the problem size increases.<br>▪ Needs uniform configuration spaces. |
| [56] | "A novel hybrid grey wolf optimizer algorithm for unmanned aerial vehicle (UAV) path planning" | Hybrid GWO | Single | ▪ Generates smooth flight routes.<br>▪ Accelerates the rate of convergence and retains the ability to explore. | ▪ The optimal value is lower than GWO, SA, and SOS.<br>▪ Execution time is higher than GWO in all cases. |

| | | | | | |
|---|---|---|---|---|---|
| [57] | "Continuous-Time Trajectory Optimization for Decentralized Multi-Robot Navigation" | DA | Multiple | ▪ Generates collision-free trajectories. ▪ Reduces jerk and time. | ▪ Robustness and scalability can fail sometimes. ▪ It has dynamic speed limits. |
| [58] | "A Self-Heuristic Ant-Based Method for Path Planning of Unmanned Aerial Vehicle in Complex 3-D Space with Dense U-Type Obstacles" | SHA | Single | ▪ The number of retreats reduced significantly. ▪ Time analysis enhanced compared to basic ACO. | ▪ Applied to static obstacles only. ▪ Actual taboo nodes are not used. |
| [59] | "A novel mission planning method for UAVs' course of action" | TDRS | Single | ▪ Generates multiple schemes automatically. ▪ Completes tasks in a shorter time. | ▪ Time optimization is essential for war scenarios. ▪ Variations in threat and utilization factors. |
| [60] | "A multi-objective pigeon-inspired optimization approach to UAV distributed flocking among obstacles" | Improved MPIO | Single | ▪ Guarantees stable and collision-free flocking. ▪ Prior environmental details and the number of UAVs are essential. | ▪ Lacks convergence analysis. ▪ Deadlocks can occur. ▪ Emergency conditions and dynamic obstacles are not tested. |
| [61] | "Application of the ACO algorithm for UAV path planning" | ACO | Single | ▪ Intermediate waypoint concept introduced for ACO. ▪ Improved fitness value. | ▪ Search space is bigger due to ACO hunting procedure. ▪ Higher computational complexity. |
| [62] | "A method of feasible trajectory planning for UAV formation based on bi-directional fast search tree" | Bi-RRT | Single | ▪ Solves the minimum efficiency of compound models in complicated environments. ▪ Yields safe and efficient formation and obstacle avoidance. | ▪ GA algorithm has a smoother path than Bi-RRT. ▪ Can move very close to an obstacle. |
| [63] | "Towards a PDE-based large-scale decentralized solution for path planning of UAVs in shared airspace" | PDE | Single | ▪ Ensures collision-free and optimal path flight safety. ▪ Proves to be computationally efficient. | ▪ Does not allow UAVs to share their trajectories during the mission. |
| [64] | "Optimized multi-UAV cooperative path planning under the complex confrontation environment" | Improved GWO | Multiple | ▪ Minimizes fuel costs and threats. ▪ Proves to be effective in cooperative path planning. | ▪ The average distance of most UAVs is greater. |
| [65] | "A constrained differential evolution algorithm to solve UAV path planning in disaster scenarios" | CDE | Single | ▪ Refines the limitations. ▪ Continues the investigations. | ▪ Used only unconstrained optimization problems. |
| [66] | "A novel reinforcement learning-based grey wolf optimizer algorithm for unmanned aerial vehicles (UAVs) path planning" | GWO | Single | ▪ Achieves effective and feasible routes smoothly. ▪ Enables each UAV to perform operations independently. | ▪ Not efficient in solving other sorts of an issue at the same time while introducing another algorithm. |

| [67] | "Synergistic path planning of multi-UAVs for air pollution detection of ships in ports" | PSO | Multiple | ▪ Detects air pollution efficiently. ▪ Guarantees reduction of ship emissions. | ▪ Does not cover air control and wind speed influences. ▪ Lacks large-scale data testing. |
|---|---|---|---|---|---|
| [68] | "An intelligent cooperative mission planning scheme of UAV swarm in uncertain dynamic environment" | HAPF ACO | Multiple | ▪ Enhances searching abilities. ▪ Executes tasks and avoids collisions and obstacles efficiently. | ▪ Aims cooperative search-attacks at homogeneous UAVs only. |
| [69] | "Path planning of multiple UAVs with online changing tasks by an ORPFOA algorithm" | ORPFOA | Multiple | ▪ Solves tasks efficiently with task preference and swapping tasks. ▪ Determines optimal paths smoothly. | ▪ Needs more reduction in running time. ▪ It has some complex computations. |
| [70] | "Path Planning for Multi-UAV Formation Rendezvous Based on Distributed Cooperative Particle Swarm Optimization" | DCPSO | Multiple | ▪ All UAVs arrived simultaneously without collision. ▪ It avoids all types of obstacles. | ▪ It cannot be used in real-time scenarios. ▪ It takes more time to avoid collisions. |
| [71] | "A Performance Study of Bio-Inspired Algorithms in Autonomous Landing of Unmanned Aerial Vehicle" | BOA MFO ABC | Single | ▪ MFO obtains the best points with minimal run time and error. ▪ Gives bearable accuracy. | ▪ Error is not optimized. |
| [72] | "UAVs path planning architecture for effective medical emergency response in future networks" | CVRP PSO ACO GA | Single | ▪ CVRP outperforms with the least runtime and minimal cost and enhanced capacities. ▪ Achieves the proper navigation. | ▪ Lacks benchmark solutions. Does not consider real-time or complex scenarios. |
| [73] | "Path planning of multiple UAVs using MMACO and DE algorithm in dynamic environment" | MMACO DE | Multiple | ▪ Increases the robustness. ▪ Preserves the global convergence speed. | ▪ In multi-colonies, one colony follows same path as basic ACO. |
| [74] | "Multi-UAV coordination control by chaotic grey wolf optimization-based distributed MPC with event-triggered strategy" | Chaotic GWO | Multiple | ▪ Gives efficiency in computations. ▪ Enhances the global search mobility convergence speed. | ▪ Stability conditions are not analyzed. ▪ Has limited communication. |
| [75] | "Collective Motion and Self-Organization of a Swarm of UAVs: A Cluster-Based Architecture" | PSO | Multiple | ▪ Gives fast connectivity and convergence. ▪ Assures stability with fewer turns. | ▪ Not implemented on hardware. ▪ Focused on a specific scenario. |
| [76] | "A Cluster-Based Hierarchical-Approach for the Path Planning of Swarm" | MMACO | Multiple | ▪ Gives superior performance. ▪ Gives an optimal path with better convergence. | ▪ Variation in the optimization costs in colonies 2 and 3 is neglected. |
| [77] | "Cooperative Path Planning of Multiple UAVs by using | MMACO CM | Multiple | ▪ Finds the optimal routes with the shortest distance. ▪ Avoids collision. | ▪ Enhances the system complexity. |

| | | | | | | |
|---|---|---|---|---|---|---|
| | Max-Min Ant Colony Optimization along with Cauchy Mutant Operator" | | | | | |
| [78] | "A multi-strategy pigeon-inspired optimization approach to active disturbance rejection control parameters tuning for vertical take-off and landing fixed-wing UAV" | MPIO | Single | ▪ Proves to be superior among all algorithms to solve multi-dimensional searching issues. ▪ It converges faster and exploits in a better way. | ▪ Altitude fluctuation is still present. ▪ Immature result after 2nd iteration. | |
| [79] | "Landing route planning method for micro drones based on hybrid optimization algorithm" | DO | Multiple | ▪ Shows stronger convergence both locally and globally. ▪ Yields better outcomes than both single algorithms. | ▪ Speeds up convergence after orthogonal learning. | |
| [80] | "Energy Efficient Neuro-Fuzzy Cluster-based Topology Construction with Metaheuristic Route Planning Algorithm for Unmanned Aerial Vehicles" | QALO | Single | ▪ Gives more energy-efficient results, more rounds, higher throughput, and lower average delay results. ▪ Selects optimal routes. | ▪ Does not manage resources optimally. | |
| [81] | "Coordinated path following control of fixed-wing unmanned aerial vehicles in wind" | CPFC | Single | ▪ Attains leaderless synchronization. ▪ Satisfies UAVs' constraints and upper bound path following errors. | ▪ Requires better simulation of the external environment and the wireless communications. | |
| [82] | "A diversified group teaching optimization algorithm with segment-based fitness strategy for unmanned aerial vehicle route planning" | GTO | Single | ▪ Gives faster convergence. ▪ Handles all the complex constrained problems. | ▪ Parameters need automatic adjustments. | |
| [83] | "Coverage path planning for multiple unmanned aerial vehicles in maritime search and rescue operations" | RSH | Multiple | ▪ Gives optimal results in a shorter time. ▪ Robust to strong wind. | ▪ Does not provide exact solutions for larger instances. | |
| [84] | "Hybrid FWPS cooperation algorithm based unmanned aerial vehicle constrained path planning" | FWPSALC | Single | ▪ Produces high and superior quality solutions. ▪ Handles constraints in a better way. | ▪ Gives poor performance for fewer number of particles or a large number of fireworks. | |
| [85] | "Safety-enhanced UAV path planning with spherical vector-based particle swarm optimization" | PSO | Single | ▪ Reduces the cost function. ▪ Gives the shortest and smoothest paths with fast convergence. | ▪ Faces premature convergence. | |

In 2019, Yang et al. [42] proposed a spatial refined voting mechanism and PSO algorithm that gave a 4D-space path planning that was collision-free and obstacle-free for multi-UAVs. Duan et al. [43] used a dynamic discrete pigeon-inspired optimization technique for search attack missions by using distributed path generation and central tasks mission. Jain et al. [44] suggested MVO and Munkres algorithms for the path planning

and coordination of multiples, it compared the results with the results of BBO and GSO and concluded that the proposed algorithm is highly efficient in reducing execution time and finding optimized path costs. Pérez-Carabaza et al. [45] worked on optimizing trajectories for UAVs that used less time in searching for targets, avoided collisions, and maintained communication. Then, there is a comparison of this MMAS-based algorithm with GA and CEO, and it yielded better results than they yield. Shao et al. [46] used comprehensively modified PSO for the path planning of UAVs. This method gave a faster and improved convergence rate and solution optimality when compared with SPSO and MGA.

Mah et al. [47] suggested a joint optimization method that gave the best secrecy performance to combat eavesdropping on the flight path and transmits power and gave superior results to the max SNR method. Bo Li et al. [48] designed an improved ACO algorithm based on the metropolis criterion and predicted three trajectory corrections schemes for collision avoidance protocols and the inscribed circle method for smoothness. Geovanni et al. [49] proposed an optimized path planning method using a meta-heuristic in the continuous 3D environment. The study also minimizes the path length in the presence of static obstacles by manipulating control inputs. Ning et al. [50] solved the task-planning issue of multi-target and multi-aircraft by proposing a two-layer mission-planning model depending on the annealing and TS algorithms. Lihua et al. [51] gave an online priority configuration algorithm for the UAV swarm flight in an environment having compounded obstacles and showed superiority in cost of energy and time in simulation results.

In 2020, Xu et al. [52] solved the LQG problem of quad-rotor UAVs by presenting a Gaussian information fusion control (GIFC) method that allowed accurate trajectory tracking and reduced the design complexity. Hu et al. [53] proposed a 3D multi-UAV cooperative velocity-aware motion planning using VeACA2D and VeACA3D. While comparing with LyCL and PALyCL, this algorithm gave higher possibilities of reaching the destination while following shorter paths and reduced time costs. Gao and li [54] considered the distributed cooperation approach formed on situation awareness consensus and its details processing method for UAV swarms. Shang et al. [55] linked a co-optimal coverage path planning method with a PSO algorithm for aerial scanning of compounded models. Qu et al. [56] evaluated a novel hybrid grey wolf optimizer algorithm with MSOS and gave better and improved results for UAV path planning in a complex environment.

Krishnan et al. [57] optimized the continuous-time trajectory by combining a decentralized algorithm with third-order dynamics that helped robots to re-plan trajectories. Zhang et al. [58] introduced an ant-based self-heuristic method for path planning of multi-UAVs. In this study, the authors used U-shaped dense complex 3D space to reduce the confusion of obstacle detection. It reduces the deadlock state with a two-stage strategy. Zhou et al. [59] utilized the multi-string chromosome genetic and cuckoo search algorithms to improve the MDLS algorithm. This improved algorithm proved that it had a better global optimization capability and diversified scheme options, and completed tasks in a shorter time as compared to the simplified MDLS. Qiu and Duan [60] developed an improved MPIO formulated on hierarchical learning behavior that gave improved distributed flocking among obstacles. Comparison with MPIO and NSGA-II showed that the improved MPIO proved to be more suitablefort handling the various-objective optimization and obstacle avoidance for UAV flocking.

Konatowski and Pawłowski [61] presented a path planning for UAVs with the help of ACO. It uses waypoints along its path with unknown parameters. The proposed work reduces the computational time and obtains the optimal route. Huang and Sun [62] detailed an approach to feasible trajectory planning formation that depends on a bi-directional fast search tree for UAVs. Radmanesh et al. [63] applied a PDE-based large-scale decentralized approach and compared it with centralized and sequential approaches to obtain collision-free and optimal path planning of multiple UAVs. Xu et al. [64] linked the

grey wolf optimizer algorithm with the PSO algorithm to achieve cooperative path planning of multi-UAVs under the threats of ground radar, missiles, and terrain. Yu et al. [65] introduced an improved constrained differential evolution algorithm that reduced the fitness functions and satisfied the three constraints, namely, height, angle, and slope of UAVs.

Later, this improved algorithm was compared with FIDE, DE variants, RankDE, CMODE, and $(\mu + \gamma) - \text{CDE}$ and proved that the proposed CDE generated more optimal paths smoothly. Qu et al. [66] used a reinforcement learning-based grey wolf optimizer algorithm. Then, compared the outcomes with the results of GWO, MGWO, EEGWO, and IGWO algorithms and concluded that the proposed RLGWO gives better, feasible, and effective path planning for UAVs. Shen et al. [67] solved the air pollution detection problem for ships in ports and evaluated a synergistic path planning of multiple UAVs. He suggested an improved PSO algorithm with a Tabu Search (TS) table, proved the efficient detection of air pollution, and ensured less emission by ships.

Zhen et al. [68] gave an improved method that is a hybrid artificial potential field with ant colony optimization (HAPF-ACO) method that executes tasks and avoids collisions and obstacles efficiently for the cooperative mission planning of fixed-wing UAVs. The results were compared with ACOAPF and PSO algorithms that proved the suggested algorithm to be highly efficient in task execution. Li et al. [69] detailed an ORPFOA algorithm that allows online changing tasks for optimal path planning of multi-UAVs for solving faster and giving higher optimization. Then, the outcomes of this suggested algorithm were compared with GWO, PSO, PIO, PSOGSA, PPPIO, and FOA. The proposed algorithm gave faster convergence and optimization than the others.

Shao et al. [70] obtained multi-UAV path planning by using the distributed cooperative PSO approach. This study presents a complex dynamic environment with a higher success rate of 0.9 compared to CCGA. Ilango and R. [71] studied Bio-inspired algorithms and analyzed their performance in the autonomous landing of UAVs. Wu et al. [72] applied a new method to UAVs that is based on consensus theory for their formation control as well as obstacle avoidance.

In 2021, recent research by Ali et al. [73] developed a multi-colonies optimization and combined MMACO and DE techniques for the cooperative path planning of many UAVs in a dynamic environment. WANG et al. [74] proposed an MPC framework along with Chaotic Grey Wolf Optimization (CGWO) and an event-triggered approach to give UAV coordination control and trajectory tracking. Ali et al. [75] used combined movement along with the reflexivity of a UAV swarm via the cluster-based technique by combining the PSO algorithm with the MAS. It showed better convergence and durability. Shafiq et al. [76] suggested a cluster-based hierarchical approach for control and path planning. It quickly finds the optimal path along with the minimal costs. Ali et al. [77] applied a hybrid algorithm of the max-min ant colony optimization algorithm with CM operators on multiple UAVs for collective path planning. It gives the optimal global solution in minimum time. He and Duan [78] considered flying, as well as touching down, issues and suggested an improved PIO for tuning the parameters of ADRC. Liang et al. [79] developed an optimal route planning for the landing of micro-UAVs using hybrid optimization algorithms with orthogonal learning.

Pustokhina et al. [80] designed clustering that is energy efficient and plans optimal routes by developing Energy Efficient Neuro-Fuzzy Cluster-based Topology Construction with the MRP technique for UAVs. Chen et al. [81] suggested a coordination strategy for fixed-wing UAVs with wind disturbances and developed a hardware-in-the-loop (HIL) simulation. Jiang et al. [82] worked on path planning for UAVs under various obstacles and proposed a diversified group teaching optimization algorithm with a segment-based fitness approach that has better global exploration ability. Cho et al. [83] gave a coverage path planning strategy with two phases for multi-UAVs that helped in searching and rescuing in maritime environment. Zhang et al. [84] presented a hybrid FWPSALC mechanism for the path planning method for UAVs that proved to be robust in searching

and handling constraints and had a better speed convergence. Phung and Ha. [85–88] developed a novel technique with spherical vector-based particle swarm optimization (SPSO) that ensures safety, feasibility, and optimal paths and gives results better than classic PSO, QPSO, θ-PSO, and various other algorithms.

## 6. Discussion

The most crucial challenge in the field of UAVs is efficient motion planning. It requires a state-of-the-art optimization method to counter issues. This research evaluates various challenges faced by UAVs and all the current designs of motion planning techniques. The recent developments discussed the results in high adaptable ability, cost and time reductions in task executions, energy efficiency, obstacles, and collision avoidance.

While reviewing various motion planning approaches, it became evident that most of the researchers preferred to use an optimization approach with nature-inspired algorithms. While discussing numerous categories of path planning strategies, it appears that hybrid algorithms give better performance. These improved and optimized algorithms overcome the limitations of numerical and analytical techniques. By analyzing the manuscript, it can be concluded that the best optimization approaches are swarm-based due to their exceptional ability to solve complex issues with their simplified approach.

## 7. Conclusions

UAVs are flying machines that possess safe and task-oriented mobility in the presence of uncertainties with the help of modified techniques and the latest technological developments. The autonomous capability of these machines is also advancing and upgrading to provide efficient flying and stable formation in dynamic environments. However, motion planning issues in UAVs are most challenging among scholars. In this article, a detailed comparative study on the motion planning issues and achievements of UAVs has been presented, along with the limitations of each article. The study also presents recent challenges in all possible categories of UAVs to highlight the importance of UAVs in our society along with their developments and state-of-the-art work performed in the last 3 years.

## 8. Future Work

There is a very bound analysis in the comparison field of motion planning and optimization algorithms that exists already and the determination of the best among them. To deploy the multiple UAV systems in a finer way, various challenges and possibilities need more exploration, as well as a reduction in exploitations. Leads for future work are to model different swarm-based intelligent optimization approaches with high accuracy and efficiency and further feasible algorithms for 3D-path planning strategies.

**Author Contributions:** Conceptualization, Z.A.A.; methodology, A.I.; software, E.H.A.; validation, E.H.A.; investigation, A.I.; data curation, E.H.A.; writing—original draft preparation, A.I. and J.J.J.; writing—review and editing, E.H.A. and A.I.; supervision, Z.A.A.; project administration, Z.A.A.; funding acquisition, J.J.J. All authors have read and agreed to the published version of the manuscript.

**Funding:** This research was supported by the European Regional Development project Green Smart Services in Developing Circular Economy SMEs (A77472).

**Data Availability Statement:** All the data are in the article.

**Conflicts of Interest:** The authors declare no conflict of interest.

**Abbreviations**

| Acronyms | Definitions |
|---|---|
| UAV | Unmanned Aerial Vehicles |
| AI | Artificial Intelligence |
| P2P | Point-to-Point |
| MAC | Medium Access Control |
| IETF | Internet Engineering Task Force |
| MAVLink | Micro Air Vehicle Link |
| NBC | Nuclear, Biological, and Chemical |
| CAS | Collision Avoidance System |
| IR | InfraRed |
| GA | Genetic algorithm |
| EP | Evolutionary Programming |
| ES | Evolutionary Strategy |
| DE | Differential Evolution |
| HS | Harmony Search |
| AIS | Artificial Immune System |
| PSO | Particle Swarm Optimization |
| BFO | Bacteria Foraging Optimization |
| CS | Cuckoo Search |
| ABC | Artificial Bee Colony |
| ACO | Ant Colony Optimization |
| CRO | Coral Reef Optimization |
| TLBO | Teaching-Learning Based Optimization |
| FA | Firefly algorithm |
| SFLA | Shuffled Frog Leaping algorithm |
| PIO | Pigeon Inspired Optimization |
| SA | Simulated Annealing |
| GSA | Gravitational Search algorithm |
| COA | Chaotic Optimization algorithm |
| IWD | Intelligent Water Drops |
| MOA | Magnetic Optimization |
| TS | Tabu Search algorithm |
| ICA | Imperialistic Competition algorithm |
| MACO | Metropolis Criterion ACO |
| MA | Munkres algorithm |
| GIFC | Gaussian information fusion control |
| DA | Decentralized algorithm |
| SHA | Self-Heuristic Ant |
| TDRS | Task Decomposition Recourse Scheduling |
| CDE | Constraint Differential Evolution |
| PDE | Partial Differential Equation |
| DCPSO | Distributed Cooperative Particle Swarm Optimization |
| DO | Dragonfly Optimization |
| QALO | Quantum Ant Lion Optimization |
| CPFC | Coordinated Path Following Control strategy |
| RSH | Randomized Search Heuristic |
| GTO | Group Teaching Optimization |
| SDCM | Swarm Distributed Cooperation Method |
| MFO | Moth Flame Optimization |
| BOA | Bat Optimization algorithm |

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
