# Peer review of "Optimization Methods Applied to Motion Planning of Unmanned Aerial Vehicles: A Review"

_drones, doi:10.3390/drones6050126_

Round 1
Reviewer 1 Report
In the manuscript, the authors introduced a comparative analysis of motion control and Path Planning of Unmanned Aerial Vehicles. Following revisions are recommended in the manuscript:
The authors should describe the related research of UAV navigation and path planning in detail.
It should be clearly mentioned in tabular form as well about what other researchers have done in other survey papers regardings UAV path planning and navigation. The authors can get some motivation from IEEE Communications Surveys & Tutorials papers (like “Caching in Vehicular Named Data Networking: Architecture, Schemes and Future Directions”).
The classification of Algorithms should have been done considering the explicit properties of the UAV path planning.
It is highly recommended to extend the work and create a comprehensive review study on UAV path planning and motion control. Instead, the authors should also avoid the inclusion of redundant discussion of irrelevant topics.
The authors should also use relatable headings and subheadings.
In Table 1 the authors should use a more relevant parameter instead of “Compared with”.
The authors should extend section 5 and should compare the algorithms in both a qualitative and quantitative manner. Section 5 is too brief.
The authors should investigate and introduce multiple evaluation parameters to differentiate the performance of the schemes introduced in the literature.
The authors should also include illustrations to clearly explain the algorithms where needed.
The writing style of the manuscript requires major revision to get some logical flow.
The authors should also read the manuscript carefully and correct the English grammar mistakes.
Author Response
The compliance reply is attached for your kind consideration.

Reviewer 2 Report
This paper provides a detailed survey study on the formation control of UAVs. The review covers a myriad of topics. Some, may not sound directly related to the title of the paper, though. Frankly, i expected to see a thorough review of the "formation control" part of the UAV and the algorithmic foundation behind it.
Also, some paragraphs are too long. My suggestion to the authors is to break them down to sub-paragraphs.
Also, it would be helpful to the reader if a list of abbreviations and/or nomenclature was added to the paper.
Another suggestion that I can make is to include some pictorial representation of the survey and the inter-connection between different topics covered in this paper. At some points, the relationship sounds a bit disjoint.
In short, providing a depth-first-type of survey might be more interesting to the research community than a breadth-first-type survey.
Author Response

(The authors gave the same response as above.)

Reviewer 3 Report
This is a survey paper regarding the formation control and the path planning of the unmanned aerial vehicles. The authors have done good job to summarize the current state-of-the-art technology and the paper is ready to be published.
Author Response
Thank you for your kind comments
Reviewer 4 Report
The work seems very disorganized. The title indicates that it is intended to be an overview of formation fligt control and path planning methods of flying robots. However, the division of methods has not been ranked and systematized in any way. A lot of important works in this area have been omitted and the period since 1999 has been really rich in works on methods of controlling robotic formations and path planning algorithms fro UAVs. What is chapter 2 on UAV challenges for? The authors have already undertaken the description of a selected field of science, which is the formation flight control of UAVs, and the should focus on it. Why are communication protocols described in Chapter 3? The extra headings in bold in the paragraphs confuse the reader. Why aren't these subsections appropriately enriched and expanded and marked as a subsection? The article should be carefully corrected and reorganized. Moreover, you should stick to the topic indicated in the title.
Author Response

(The authors gave the same response as above.)

Round 2
Reviewer 4 Report
The article has been improved since the previous version. I only suggest adding these positions to the reference articles:
Suo, W.; Wang, M.; Zhang, D.; Qu, Z.; Yu, L. Formation Control Technology of Fixed-Wing UAV Swarm Based on Distributed Ad Hoc Network. Appl. Sci. 2022, 12, 535. https://doi.org/10.3390/app12020535
Zong, Q.; Wang, D.D.; Shao, S.K.; Zhang, B.; Han, Y. Research status and development of multi UAV coordinated formation flight control. J. Harbin Inst. Technol. 2017, 49, 1–14
Ambroziak, L.; Ciężkowski, M. Virtual Electric Dipole Field Applied to Autonomous Formation Flight Control of Unmanned Aerial Vehicles. Sensors 2021, 21, 4540. https://doi.org/10.3390/s21134540